# A Phenome-Wide Association Study (PheWAS) of COVID-19 Outcomes by Race Using the Electronic Health Records Data in Michigan Medicine

**DOI:** 10.3390/jcm10071351

**Published:** 2021-03-25

**Authors:** Maxwell Salvatore, Tian Gu, Jasmine A. Mack, Swaraaj Prabhu Sankar, Snehal Patil, Thomas S. Valley, Karandeep Singh, Brahmajee K. Nallamothu, Sachin Kheterpal, Lynda Lisabeth, Lars G. Fritsche, Bhramar Mukherjee

**Affiliations:** 1Department of Biostatistics, University of Michigan School of Public Health, 1415 Washington Heights, Ann Arbor, MI 48109, USA; mmsalva@umich.edu (M.S.); gutian@umich.edu (T.G.); jasamack@umich.edu (J.A.M.); snehal@umich.edu (S.P.); larsf@umich.edu (L.G.F.); 2Center for Precision Health Data Science, University of Michigan, Ann Arbor, MI 48109, USA; swarsank@umich.edu; 3Department of Epidemiology, University of Michigan School of Public Health, Ann Arbor, MI 48109, USA; llisabet@umich.edu; 4Rogel Cancer Center, Michigan Medicine, Ann Arbor, MI 48109, USA; 5Data Office for Clinical and Translational Research, University of Michigan, Ann Arbor, MI 41809, USA; 6Division of Pulmonary and Critical Care Medicine, University of Michigan Medicine, Ann Arbor, MI 48109, USA; valleyt@med.umich.edu; 7Department of Internal Medicine, Michigan Medicine, Ann Arbor, MI 48109, USA; bnallamo@med.umich.edu; 8Institute for Healthcare Policy and Innovation, University of Michigan, Ann Arbor, MI 48109, USA; kdpsingh@med.umich.edu (K.S.); sachinkh@med.umich.edu (S.K.); 9Department of Learning Health Sciences, University of Michigan, Ann Arbor, MI 48109, USA; 10Division of Cardiovascular Medicine, Michigan Medicine, Ann Arbor, MI 48109, USA; 11Department of Anesthesiology, Michigan Medicine, Ann Arbor, MI 48109, USA; 12Center for Statistical Genetics, University of Michigan School of Public Health, Ann Arbor, MI 48109, USA

**Keywords:** biobank, health disparities, EHR, phenome, odds ratio, risk profile

## Abstract

Background: We performed a phenome-wide association study to identify pre-existing conditions related to Coronavirus disease 2019 (COVID-19) prognosis across the medical phenome and how they vary by race. Methods: The study is comprised of 53,853 patients who were tested/diagnosed for COVID-19 between 10 March and 2 September 2020 at a large academic medical center. Results: Pre-existing conditions strongly associated with hospitalization were renal failure, pulmonary heart disease, and respiratory failure. Hematopoietic conditions were associated with intensive care unit (ICU) admission/mortality and mental disorders were associated with mortality in non-Hispanic Whites. Circulatory system and genitourinary conditions were associated with ICU admission/mortality in non-Hispanic Blacks. Conclusions: Understanding pre-existing clinical diagnoses related to COVID-19 outcomes informs the need for targeted screening to support specific vulnerable populations to improve disease prevention and healthcare delivery.

## 1. Introduction

The emergence of electronic health records (EHR) and rise of EHR-linked biobanks have made it possible for researchers to explore omics-based relationships agnostically on a large scale instead of targeted hypothesis testing. Introduced by Denny et al. in 2010, a phenome-wide association study (PheWAS) is an omnibus scan to identify gene–disease associations across the medical phenome [1]. PheWAS typically associates a genetic variant (G) with hundreds of disease codes ($D_{j},\;j\;=\;1,\ldots ,J$) with association models of the structure $logit\left(P\left(D_{j}|G,\;Confounders)\right)\right)\;=\;\beta_{0j}+\beta_{Gj}G+\beta_{C}Confounders$. Due to computational advances and development of widely available analytic frameworks [2,3,4,5,6], PheWAS is now relatively easy to implement. The main goal of a PheWAS is to replicate known gene–disease relationships and to search for hidden and unanticipated associations (for example, Li et al. found that there is a strong negative association between the tag single nucleotide polymorphism (SNP) for blood group O antigen and arm impedance) [2,7,8,9,10].

As of 15 January 2021, there were 23,759,743 confirmed COVID-19 cases in the US [11], representing approximately 25% of all global cases. Because COVID-19 is a respiratory disease and produces flu-like symptoms, the testing strategies in the US initially focused on those with symptoms, the elderly, and those with pre-existing conditions [12]—i.e., populations who are at risk of severe disease and complications. Only a handful of pre-existing comorbidities are known to be associated with experiencing adverse COVID-19-related outcomes. These pre-existing conditions include liver, kidney, heart, and respiratory disease [2,13,14,15,16,17].

There has been a remarkable surge within the academic and medical communities to conduct rapid research on COVID-19 [18]. There have been a number of studies examining differences across racial groups for an ensemble of COVID-19-associated conditions and outcomes in US patient cohorts [13,19,20,21,22,23,24,25]. Instead of a hypothesis-driven approach a priori restricted to certain disease categories, this study applied an agnostic *disease–disease* PheWAS framework to COVID-19 outcomes in a cohort of 53,853 patients who were tested or diagnosed with COVID-19 at a large academic medical center. We looked at correlates of disease prognosis among all COVID-19 patients as well as separately among non-Hispanic White (White) and non-Hispanic Black/African American (Black) patients. The primary objective of this study was to agnostically identify pre-existing conditions present in an individual’s medical record that may be associated with hospitalization, intensive care unit (ICU) admission, and mortality. We also present the results from race-stratified susceptibility PheWAS to predict who tests positive for COVID-19 in the Appendix A. Our reason to downplay the outcome of who gets COVID-19 or who tests positive for COVID-19 is due to the prioritized testing strategy that makes this tested sample highly non-representative of the population. A naïve comparison of the positive versus negative test results is highly biased [13]. However, conditional on testing or being diagnosed positive, downstream prognostic outcomes are less prone to such selection biases and we primarily focus on these outcomes.

## 2. Materials and Methods

### 2.1. Study Design

#### COVID-19 Cohort

We extracted the EHR for patients who were tested for COVID-19 at the University of Michigan Health System, also known as Michigan Medicine (MM), from 10 March 2020 to 2 September 2020. A total of 53,260 patients (98.9%) who were tested at MM and 593 patients (1.1%) who were treated for COVID-19 in MM, but tested elsewhere, constituted our initial study cohort of 53,853 patients, of whom 2582 tested positive. Our analytic cohort was restricted to those individuals on whom we have EHR data for at least 14 days prior to the first COVID-19 test. This restriction is used to eliminate symptoms which may indicate manifestation of underlying COVID-19 disease or symptoms, whereas our goal was to search for truly “pre-existing” conditions prior to COVID-19 testing/diagnosis. Our resulting analytic cohort comprised 47,862 tested/diagnosed patients, of whom 2133 tested positive. Since the testing protocol in MM [26] focused on prioritized testing based on symptoms, exposure, occupation and other patient level factors, this is a non-representative sample of the population. Study protocols were reviewed and approved by the University of Michigan Medical School Institutional Review Board (IRB ID HUM00180294).

### 2.2. Data Processing

#### 2.2.1. Classifying Patients Who Were Still in Hospital and ICU to Define COVID-19 Outcomes

We categorized COVID-19-positive patients into non-hospitalized, hospitalized (includes ICU stays), and hospitalized with ICU stay based on the admission and discharge data. A total of 22 patients were still admitted in the hospital at the time of data extraction (17 had at least one ICU stay and five had no ICU stay). 

#### 2.2.2. Generation of the Medical Phenome

We constructed the medical phenome by extracting available International Classification of Diseases (ICD; ninth and tenth editions) codes from EHR and grouping them into 1813 traits using the PheWAS R package (as described in [1]). Each of these traits (PheWAS codes) was coded as a binary risk factor (present/absent) and used as a predictor in the association models with COVID-19 outcomes. As mentioned before, to differentiate *pre-existing* conditions from phenotypes related to COVID-19 testing/treatment, we applied a 14-day-prior restriction on the tested cohort by removing diagnoses that first appeared within the 14 days before the first test or diagnosis date, whichever was earlier. The analyses in this study were restricted to 1363 traits that appeared in the EHR 14-day-prior of at least ten COVID-19-positive patients. While the PheWAS is performed on PheWAS codes, one can view the mapping of ICD-to-PheWAS code relationships on this website: https://prsweb.sph.umich.edu:8443/phecodeData/searchPhecode (accessed on 22 September 2020).

#### 2.2.3. Description of Variables

A summary data dictionary is available with the source and definition of each variable used in our analysis (Appendix A).

### 2.3. Statistical Analysis

We performed PheWAS to identify predictors of three COVID-19 prognostic outcomes in this study (detailed definition in Appendix A), among those who were diagnosed/tested positive, comparing:(i)those who were hospitalized with those who were not;(ii)those who were admitted to ICU or died with those who were not;(iii)those who died with those who were alive at the time of data extraction.

We also present results from the susceptibility PheWAS (comparing those who were diagnosed with COVID-19 with those who were not tested at all [matched controls]) in the Appendix A.

All COVID-19 outcomes of interest are binary; thus, logistic regression was our primary tool for association analysis. All logistic regression models were of the following form:(1)$$\mathrm{logit}\;p{(Y}_{\mathrm{COVID}}\;=\;1|\mathrm{Covariates},{\;\mathrm{PheCode}}_{j})\;={\;\beta }_{0}+\beta_{\mathrm{Cov}}^{⊤}\mathrm{Covariates}+\beta_{j}I\left[{\mathrm{Phecode}}_{j}\;=\;1\right]$$j = 1, …, 1363. Here $Y_{\mathrm{COVID}}\;$is various COVID-19-related outcomes under consideration (e.g., COVID-19 hospitalization, ICU admission, and mortality). The Firth correction was used to address potential separation issues in logistic regression models. For all models, adjusted odds ratios (OR), 95% Wald-type confidence interval and *p*-values were presented [27,28,29]. Full models were adjusted for age, sex, race, and the neighborhood deprivation index (NDI). The NDI is defined by US census tract (corresponding to the residential address available in each patient’s EHR) for the year 2010 and are from the National Neighborhood Data Archive (NaNDA) [30]. PheWAS adjusting for an additional comorbidity score (indicating whether the patient was diagnosed with conditions across seven disease categories associated with COVID-19 susceptibility and adverse outcomes: respiratory, circulatory, any cancer, type II diabetes, kidney, liver, and autoimmune; ranges from 0 to 7) is included as a sensitivity analysis on our accompanying website: https://cphds.sph.umich.edu/covidphewas/ (accessed on 10 March 2021). Results for PheWAS analysis are easier to visualize when −log_10_(*p*-values) corresponding to each of the 1363 tests are plotted against the disease codes grouped into disease categories. We use this visualization tool to present our analysis while all detailed summary results are available at the website above and in the online supplement.

#### Race-Stratified Analysis

Since the prognostic factors could potentially be different across races, we repeated the entire analysis stratified by race. We restricted our attention to Whites and Blacks due to limitations of sample size for other racial groups. Appendix A contains descriptive statistics stratified by race. We checked for the equality of the log(OR) corresponding to Whites and Blacks through a Wald test for the difference of the log(OR). A conservative Bonferroni multiple testing correction was implemented to conclude statistically significant results (*p* = 0.05/number of tests in analysis), and *p* < 0.05 was used as a threshold for suggestive traits.

## 3. Results

There were 53,853 patients who were either tested for or diagnosed with COVID-19 who were eligible for inclusion in this study. Of those eligible for inclusion, our study population comprised 47,862 individuals (n_tested_ = 47,862 [n_positive_ = 2133]) who had available ICD code data after applying the 14-day-prior to testing restriction to the EHR. Furthermore, a total of 1813 qualified ICD-code-based phenotypes, referred to as PheWAS codes, were initially screened, of which 1363 had at least 10 occurrences in our COVID-19-positive cohort and were included in the analysis.

Of those 53,853 who were tested for COVID-19, 44.2% (23,814) were males and the median age was 47 years. The majority were White (72.4% (38,977)), while 10.7% were Black (5763). We note that the Black cohort is both younger and more female than the White cohort, a trend that also appears in the tested and hospitalized cohorts (Appendix A). Similarly, Blacks tend to have more autoimmune disease, kidney disease, type 2 diabetes, and circulatory disease diagnoses, while Whites tend to have more cancer diagnoses (note that our definition of cancer includes skin cancer) in our tested cohort (Appendix A). Out of the study cohort, 4.8% (2582) were tested positive (Table 1). Among the 2582 positive patients, 54.6% (1411) were White, 25.0% (646) were Black, 27.8% (719) were hospitalized, 14.6% (377) were admitted to ICU and 5.0% (129) died. A flowchart describing the sample sizes of the overall cohort and race-specific cohorts by COVID-19 outcome is included in Appendix A.

### 3.1. Phenome-Wide Comorbidity Association Analysis

The association results for the top 50 traits from the comorbidity PheWAS can be found in Appendix A for the full cohort, Whites, and Blacks, side-by-side. Interactive versions of the PheWAS plots are online at https://cphds.sph.umich.edu/covidphewas/ (accessed on 10 March 2021). This online resource also provides tables with the adjusted odds ratios, 95% confidence intervals, *p*-values, and counts of occurrence in cases and controls for all traits included in the PheWAS performed.

#### 3.1.1. Full Cohort Prognostic Associations

As the disease outcome progresses (from hospitalized to ICU, and to deceased), stronger associations with circulatory system, genitourinary (renal diseases in particular) and respiratory diseases were observed. Forty-four traits, including 12 circulatory system and 11 respiratory diseases, were phenome-wide significantly associated with hospitalization, as well as an additional 263 suggestive traits under the threshold of *p* < 0.05 (Figure 1A)—respiratory failure, insufficiency, arrest (*p* = 3.98 × 10^−20^), acute renal failure (*p* = 6.31 × 10^−13^), viral pneumonia (*p* = 2.51 × 10^−11^), and acid-base balance disorder (*p* = 2.40 × 10^−10^). Moreover, 58 phenome-wide significant hits (e.g., respiratory failure, insufficiency, arrest [*p* = 1.58 × 10^−15^], acid-base balance disorder [*p* = 3.98 × 10^−14^], and hypotension [*p* = 1.58 × 10^−11^]) as well as 286 suggestive hits were noted for association with ICU admission/mortality (Figure 2A), including 77 circulatory system, 36 endocrine/metabolic, 35 genitourinary, and 31 respiratory diseases. There were 22 phenome-wide significant traits associated with COVID-19 mortality (Figure 3A), along with an additional 227 suggestive traits under the threshold *p* < 0.05. In addition to 64 circulatory system and 31 endocrine/metabolic diseases, 23 mental disorders stood out as the third largest disease group associated with mortality, including delirium due to conditions classified elsewhere (*p* = 9.33 × 10^−7^), memory loss (*p* = 3.98 × 10^−4^) and aphasia (*p* = 5.37 × 10^−4^).

#### 3.1.2. Race-Stratified Prognostic Associations

Among White patients, we identified 23 traits phenome-wide significantly associated with hospitalization (e.g., respiratory failure, insufficiency, arrest [*p* = 5.25 × 10^−9^], acute renal failure [*p* = 5.25 × 10^−9^], and electrolyte imbalance [*p* = 1.51 × 10^−8^] Figure 1B), as well as 239 suggestive traits, including 54 circulatory system, 30 respiratory, 29 endocrine/metabolic, and 21 genitourinary diseases. Thirty-two phenome-wide significant traits (e.g., electrolyte imbalance [*p* = 2.63 × 10^−8^], pulmonary collapse, interstitial and compensatory emphysema [*p* = 8.91 × 10^−8^] and hypotension [*p* = 3.63 × 10^−7^]) and 239 suggestive traits were associated with ICU admission/mortality (Figure 2B), including 60 circulatory system, 27 respiratory, 27 digestive, and 23 hematopoietic diseases. One phenome-wide significant trait (elevated white blood cell count [*p* = 3.55 × 10^−5^]) and 130 suggestive traits were associated with COVID-19 mortality (Figure 3B), including 18 circulatory system, 17 endocrine/metabolic, 16 mental disorders, 14 genitourinary diseases such as osteomyelitis (*p* = 1.74 × 10^−4^), neurological disorder (*p* = 4.37 × 10^−4^) and aphasia (*p* = 4.57 × 10^−4^).

Among Black patients, two phenome-wide significant traits were detected (respiratory failure, insufficiency, arrest [*p* = 2.63 × 10^−8^], respiratory failure [*p* = 4.37 × 10^−7^]) along with 89 traits nominally associated with hospitalization (Figure 1C), including 17 circulatory, 15 genitourinary, and 14 respiratory diseases. Eleven phenome-wide significant traits (e.g., respiratory failure, insufficiency, arrest [*p* = 1.48 × 10^−8^], acid-base balance disorder [*p* = 4.57 × 10^−6^], hypotension [*p* = 4.37 × 10^−5^]) and 119 suggestive traits were associated with ICU admission/mortality, including 33 circulatory, 26 genitourinary, and 17 endocrine/metabolic diseases (Figure 2C). Six phenome-wide significant traits (e.g., empyema and pneumothorax [*p* = 3.98 × 10^−5^], hyperosmolality and/or hypernatremia [*p* = 5.37× 10^−5^], atrial fibrillation [*p* = 5.62 × 10^−5^]) and 105 suggestive traits were associated with mortality, including 34 circulatory, 24 endocrine/metabolic, and 12 genitourinary diseases. As shown in Figure 3B,C, the strength of association between circulatory system disorders and COVID-19 mortality was higher in Black patients compared to White. Similarly, we observe a higher prevalence of genitourinary diseases in Blacks associated with COVID-19 mortality such as stage I or II chronic kidney disease (*p* = 2.34 × 10^−4^) compared to Whites. 

When comparing the effect sizes of the top 50 hits significant in the full cohort, stratified by racial groups, we found—as expected—no significant differences in the effect sizes (though there are numerical differences). These traits exhibited consistent risks among races for hospitalization (Appendix A) and for ICU admission/mortality (Appendix A). 

#### 3.1.3. Differences between Blacks and Whites across the Phenome

When comparing the estimated effect sizes for the hospitalization outcome across the phenome between Blacks and Whites, we found 35 pre-existing traits (including 10 digestive, 7 endocrine/metabolic, 6 circulatory system, and 5 respiratory conditions) where the effect estimates differed across Blacks and Whites at a *p*-value less than 0.05 (Appendix A). In all cases, there was a stronger, positive association with the trait and COVID-19-related hospitalization among Whites compared to Blacks. For the ICU admission outcome, we found 31 such pre-existing traits (including 6 neoplasm, 4 hematopoietic, 4 respiratory, and 4 digestive conditions) where the effect estimates differed (*p* for difference < 0.05) (Appendix A). Similarly, there was a stronger, positive association with the trait and COVID-19-related ICU admission among Whites compared to Blacks, except for delirium due to conditions classified elsewhere (phecode 290.2), chronic pain syndrome (phecode 355.1), disorders of lacrimal system (phecode 375), and degeneration of intervertebral disc (phecode 722.6), where the opposite was true. For the COVID-19-related death outcome, we found 8 such conditions, including 3 circulatory system and 2 respiratory conditions (Appendix A). There was a strong, positive association with the trait and COVID-19-related death among Blacks compared to Whites for 6 of the traits, while the opposite was true for decreased white blood cell count (phecode 288.1) and cervicalgia (phecode 761).

#### 3.1.4. Summary Takeaways

In all cohorts, as the disease progressed to increasingly severe prognosis, the associated phenotypes concentrated in circulatory heart diseases and renal diseases (Figure 4); pre-existing *cardiovascular system problems*, and chronic diseases such as *chronic pulmonary heart disease* and *chronic renal failure* appeared to be associated with poor prognosis, while mental disorders constituted the third largest category associated with COVID-19 mortality behind circulatory system and endocrine/metabolic diseases. When comparing the top 50 traits between Whites and Blacks, acidosis, pulmonary, acute/chronic renal diseases showed an association with hospitalization and ICU admission/mortality in both races, while acute renal consistently stood out as well as in mortality (Table 2). Phenome-wide significant (for the overall cohort) effect estimates for parent phecodes and corresponding confidence intervals for phenome-wide significant traits by outcome by cohort are present in forest plots of parent phecodes (Appendix A) and child phecodes (Appendix A). While no effect differences were significant at Bonferroni-corrected *p*-value thresholds, we observed 35, 31, and 8 nominally significant (*p* < 0.05) effect differences for Blacks compared to Whites for hospitalization (Appendix A), ICU/admission (Appendix A), and death (Appendix A), respectively. A description of the susceptibility outcome results and corresponding PheWAS plots (Appendix A) are in the Appendix A. Respiratory, endocrine/metabolic, and circulatory system conditions were associated with COVID-19 susceptibility overall, while mental disorders appear overrepresented among Whites. 

These results enable us to understand the risk profiles that are associated with poor COVID-19 prognosis. It will be interesting to study the association of these pre-existing conditions with post-covid acute complications or “long COVID” syndrome [31,32]. 

## 4. Discussion

Using data from a cohort of tested/diagnosed COVID-19 patients at MM, we performed what we believe is the first PheWAS looking at multiple COVID-19 outcomes stratified by race. A recently published PheWAS by Oetjens et al. found 21 phenome-wide significant traits—including six kidney (e.g., end stage renal disease or stage 5 CKD [OR = 11.07, *p* = 1.96 × 10^−8^]), six cardiovascular (e.g., congestive heart failure [OR = 3.8, *p* = 3.24 × 10^−5^]), five respiratory (e.g., chronic airway obstruction [OR = 2.54, *p* = 3.71 × 10^−5^]), and three metabolic (e.g., type 2 diabetes [OR = 1.80, *p* = 7.51 × 10^−5^]) phenotypes—associated with COVID-19-related hospitalization, which is consistent with our results [14]. This technique allowed us to explore and identify potentially associated conditions across the medical phenome that are associated with susceptibility, hospitalization, ICU admission or mortality. Our results yield many previously known or plausibly associated phenotypes with increasingly severe prognosis such as pulmonary heart disease, respiratory failure and type 2 diabetes. Our stratified analysis showed that kidney disease (e.g., renal failure OR = 1.34, *p* = 4.90 × 10^−7^ among Whites vs. OR = 1.12, *p* = 2.88 × 10^−4^ among Blacks for ICU admission) and hematopoietic conditions (e.g., anemia of chronic disease OR = 1.39, *p* = 7.59 × 10^−5^ among Whites vs. OR = 1.19, *p* = 4.68 × 10^−3^ among Blacks for ICU admission) appear to be associated with more severe outcomes among Whites, while respiratory (e.g., pulmonary insufficiency or respiratory failure following trauma OR = 2.75, *p* = 4.68 × 10^−6^ among Blacks vs. OR = 1.70, *p* = 6.92 × 10^−3^ among Whites for death) and circulatory system (e.g., atrial fibrillation OR = 2.26, *p* = 5.62 × 10^−5^ among Blacks vs. OR = 1.90, *p* = 2.15 × 10^−1^ among Whites for death) conditions are more strongly associated with severe outcomes among Blacks. Table 2 shows that the disease categories that comprise the top 20 hits by prognostic outcome by race are different (with the caveat that for ICU admission and mortality these hits are largely suggestive due to limited power). Our results can inform targeted prevention across racial groups, which includes increased testing and encouraging self-isolation from household members with specific disease profiles along with education of enhanced public health prevention guidelines.

There are several limitations to this analysis. First, there is the agnostic nature of PheWAS, which can identify potentially spurious associations. While we feel that many of the top traits have been highlighted elsewhere and are biologically plausible, there is currently no process in place for rapidly discerning potentially novel from spurious associations [33] beyond extensive manual review and follow-up research, particularly for a novel disease. Second, many of the issues with utilizing EHR data for research purposes also apply here, including inaccurate data from billing codes [34] and failure of physicians to report/record problems [35]. Third, the sample size for a PheWAS is still rather small to be able to identify statistically significant associations—particularly for mortality. Moreover, we did not distinguish between transfer patients (i.e., those who were diagnosed elsewhere and transferred to MM for treatment), who may have been sicker patients than the cohort diagnosed at MM. However, given that this is an emerging and novel disease, we feel it is important to identify suggestive associations so that future research and clinicians can potentially consider other conditions outside those that have been previously identified—namely, pulmonary and cardiovascular conditions—and to inspire follow-up studies in larger cohorts. For example, OpenSAFELY, a platform including the primary care records of 17,728,392 adults in England (covering 40% of all patients) [16], shares many of our full-cohort conclusions (with consistent effect sizes), but has not published race-stratified PheWAS results. Finally, our analysis is scanning through each phenotype one at a time, though they occur in a correlated and interactive manner. A richer multivariate model needs to be constructed with more complex features.

While potentially relevant, we did not explore past medication data in our analyses because the available EHR data did not provide comprehensive medication coverage but predominantly medication orders and administrations for hospitalized patients. Other population-based cohorts with prescription registries, as exemplified by Butt and colleagues [36], might be better suited for such an exploration.

## 5. Conclusions

This work contributes to a new area of COVID-19 research that rigorously examines racial differences in disease prognosis with pre-existing conditions captured across the medical phenome. Moreover, we incorporated a census tract-level socioeconomic status (SES) covariate, which is important to consider when comparing races [37]. We found several potentially novel diseases unexpectedly associated with different outcomes in the course of COVID-19 progression and that some disease profiles differ by race. For example, we provide additional evidence on the previously reported concern that patients with mental health disorders are at higher risk of infection and experience barriers in seeking treatment leading to poor prognosis [38]. We hope this exploratory effort will inspire hypothesis generation for future research that might result in targeted prevention and care as we are still combatting this pandemic. In this spirit, we have made all PheWAS results available for exploration here: https://cphds.sph.umich.edu/covidphewas/ (accessed on 10 March 2021). We hope the summary data and the phenomic landscape for COVID-19 will help future replication and meta-analysis efforts.

## Figures and Tables

**Figure 1 jcm-10-01351-f001:**
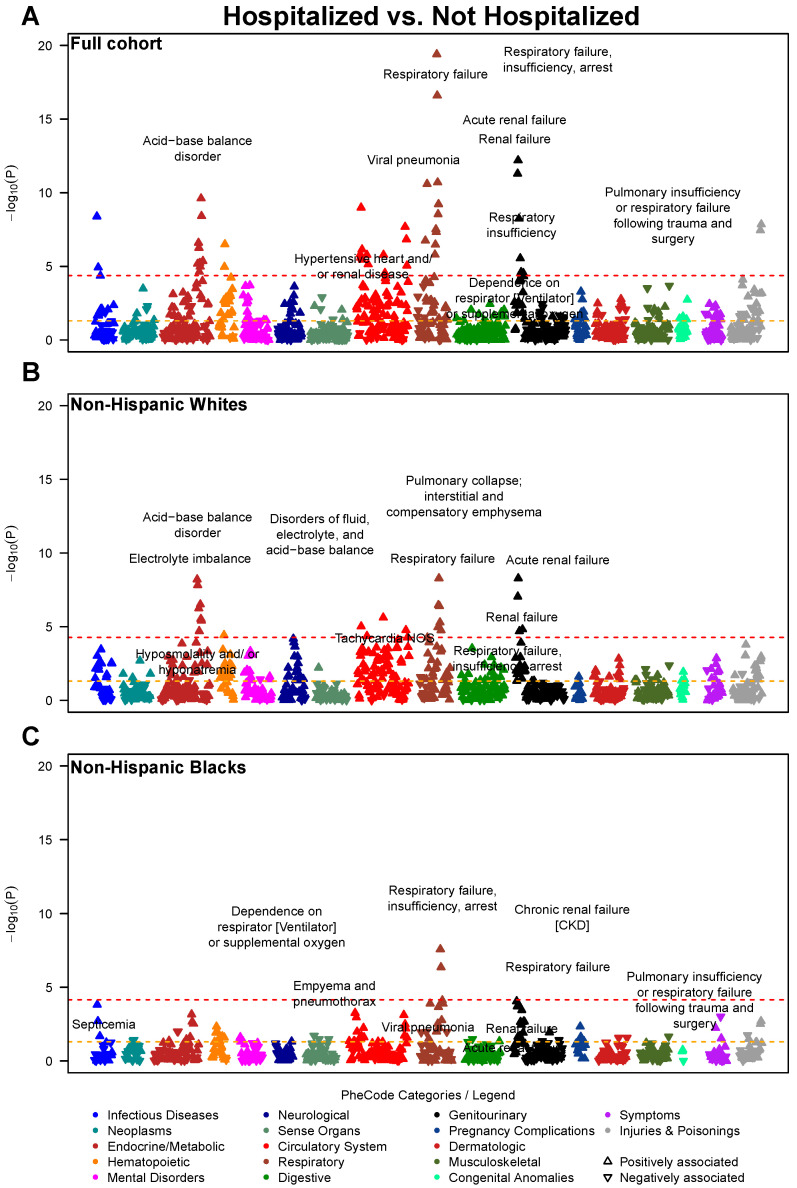
Manhattan plot showing the phenome-wide association between disease conditions and hospitalization for COVID-19. Models are adjusted for age, sex, race (full cohort only), and three census tract-level socioeconomic indicators: proportion with less than high school education, proportion unemployed, and proportion with annual income below the federal poverty level. The *x*-axis are individual disease codes, color-coded by their corresponding disease category as described in the shared legend. The *y*-axis represents the −log_10_ transformed *p*-value of the association. The dashed, horizontal lines represent the *p* = 0.05 (in orange) and the Bonferroni corrected *p*-value (0.05/number of tests; in red). Each point is represented by either an upward triangle indicating a positive association or a downward triangle indicating a negative association. (**A**): Full cohort, (**B**): Restricted to Whites, (**C**): Restricted to Blacks

**Figure 2 jcm-10-01351-f002:**
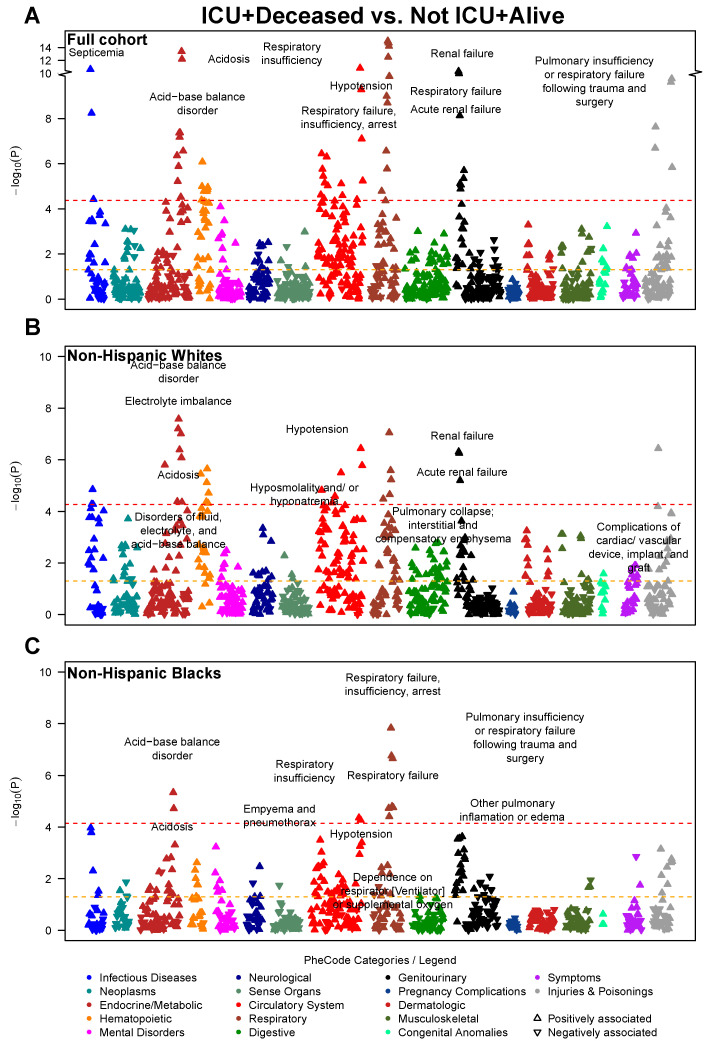
Manhattan plot showing the phenome-wide association between disease conditions and ICU admission for COVID-19. Models are adjusted for age, sex, race (full cohort only), and three census tract-level socioeconomic indicators: proportion with less than high school education, proportion unemployed, and proportion with annual income below the federal poverty level. The *x*-axis are individual disease codes, color-coded by their corresponding disease category as described in the shared legend. The *y*-axis represents the −log_10_ transformed *p*-value of the association. The dashed, horizontal lines represent the *p* = 0.05 (in orange) and the Bonferroni corrected *p*-value (0.05/number of tests; in red). Each point is represented by either an upward triangle indicating a positive association or a downward triangle indicating a negative association. (**A**): Full cohort, (**B**): Restricted to Whites, (**C**): Restricted to Blacks

**Figure 3 jcm-10-01351-f003:**
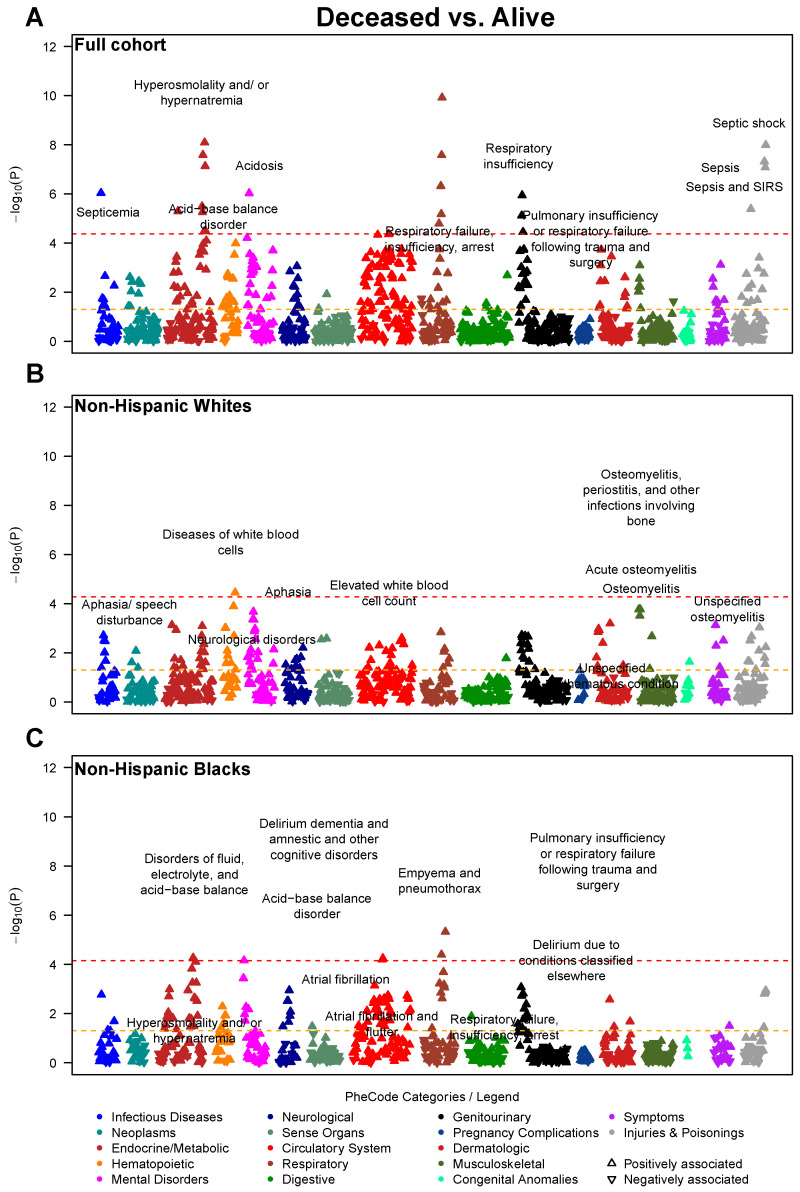
Manhattan plot showing the phenome-wide association between disease conditions and prognostic outcomes for COVID-19. Models are adjusted for age, sex, race (full cohort only), and three census tract-level socioeconomic indicators: proportion with less than high school education, proportion unemployed, and proportion with annual income below the federal poverty level. The *x*-axis are individual disease codes, color-coded by their corresponding disease category as described in the shared legend. The *y*-axis represents the −log_10_ transformed *p*-value of the association. The dashed, horizontal lines represent the *p* = 0.05 (in orange) and the Bonferroni corrected *p*-value (0.05/number of tests; in red). Each point is represented by either an upward triangle indicating a positive association or a downward triangle indicating a negative association. (**A**): Full cohort, (**B**): Restricted to Whites, (**C**): Restricted to Blacks

**Figure 4 jcm-10-01351-f004:**
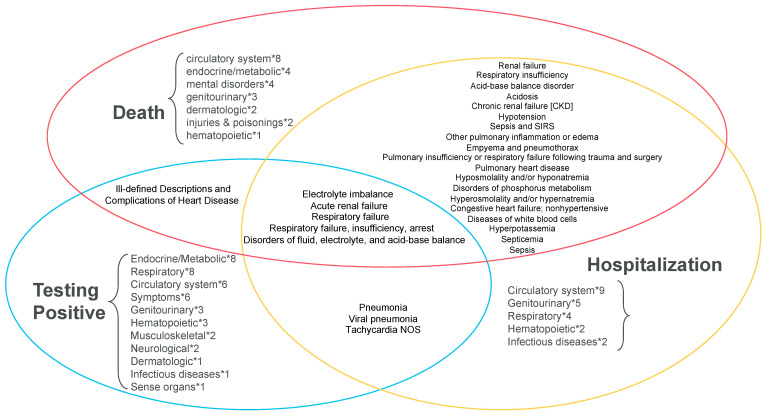
Venn diagrams of the top 50 traits. Each circle represents the top 50 hits from the full cohort PheWAS. Traits shared across PheWAS are stated, while the corresponding number of traits within a given disease category that are unique to that PheWAS are also provided. Abbreviations: NOS, not otherwise specified; SIRS, systemic inflammatory response syndrome

**Table 1 jcm-10-01351-t001:** Descriptive Characteristics of the COVID-19 Tested/Diagnosed cohort at Michigan Medicine (10 March–2 September 2020).

	Individuals, No. (%) ^a^					
	Tested for COVID-19					
			Positive Results			
	Overall	Negative Results	Overall	Hospitalized	ICU	Deceased
Variable	(*n* = 53,853)	(*n* = 51,271)	(*n* = 2582)	(*n* = 719)	(*n* = 377)	(*n* = 129)
Age, y						
Mean (SD)	44.8 (23.1)	44.7 (23.2)	47.4 (20)	58.5 (17.6)	58.6 (17.5)	69 (14.3)
Median (IQR)	47 (38)	46 (38)	49 (31)	61 (23)	61 (22)	71 (22)
<18	6895 (12.8)	6768 (13.2)	127 (4.9)	14 (1.9)	10 (2.7)	0 (0)
[18,35)	12,652 (23.5)	12,017 (23.4)	635 (24.6)	65 (9)	33 (8.8)	3 (2.3)
[35,50)	9273 (17.2)	8697 (17)	576 (22.3)	125 (17.4)	56 (14.9)	11 (8.5)
[50,65)	12,116 (22.5)	11,440 (22.3)	676 (26.2)	224 (31.2)	120 (31.8)	33 (25.6)
[65,80)	10,257 (19)	9825 (19.2)	432 (16.7)	209 (29.1)	124 (32.9)	43 (33.3)
≥80	2660 (4.9)	2524 (4.9)	136 (5.3)	82 (11.4)	34 (9)	39 (30.2)
Male Gender	23,814 (44.2)	22,651 (44.2)	1163 (45)	403 (56.1)	233 (61.8)	80 (62)
Primary Care in MM	31,357 (58.2)	29,969 (58.5)	1388 (53.8)	253 (35.2)	128 (34)	35 (27.1)
BMI						
Mean (SD)	29.1 (7.6)	29.1 (7.6)	30.9 (8.4)	32.6 (10.1)	32.9 (11.5)	31.3 (6.9)
<18.5	826 (1.9)	804 (2)	22 (1)	9 (1.3)	4 (1.1)	1 (0.8)
[18.5,25)	12,857 (29.7)	12,357 (30)	500 (22.9)	102 (14.9)	61 (16.9)	17 (13.7)
[25,30)	13,371 (30.8)	12,723 (30.9)	648 (29.7)	211 (30.9)	110 (30.5)	45 (36.3)
≥30	16,291 (37.6)	15,281 (37.1)	1010 (46.3)	361 (52.9)	186 (51.5)	61 (49.2)
Smoking Status						
Never	31,041 (63.2)	29,549 (63)	1492 (68.7)	368 (60.2)	159 (54.6)	30 (39)
Past	13,725 (28)	13,145 (28)	580 (26.7)	219 (35.8)	120 (41.2)	44 (57.1)
Current	4314 (8.8)	4215 (9)	99 (4.6)	24 (3.9)	12 (4.1)	3 (3.9)
Ever	18,039 (36.8)	17,360 (37)	679 (31.3)	243 (39.8)	132 (45.4)	47 (61)
Alcohol consumption	25,894 (68.4)	24,768 (68.6)	1126 (66.2)	261 (63.2)	128 (63.7)	35 (61.4)
Race/ethnicity						
White	38,977 (72.4)	37,566 (73.3)	1411 (54.6)	326 (45.3)	172 (45.6)	56 (43.4)
Black	5763 (10.7)	5117 (10)	646 (25)	265 (36.9)	139 (36.9)	42 (32.6)
Other ^b^	4869 (9)	4616 (9)	253 (9.8)	63 (8.8)	21 (5.6)	6 (4.7)
Unknown ^c^	4244 (7.9)	3972 (7.7)	272 (10.5)	65 (9)	45 (11.9)	25 (19.4)
NDI, mean (SD)	0.1 (0.07)	0.1 (0.07)	0.12 (0.09)	0.15 (0.1)	0.16 (0.11)	0.16 (0.11)
Population density persons/mile^2^	2375.8 (2422.1)	2343.2 (2412.8)	2997.3 (2512.8)	3658.7 (2635)	3826.4 (2675.2)	4128.4 (2770.3)
Respiratory Diseases	34,471 (72)	32,850 (71.8)	1621 (76)	399 (79.6)	205 (81.7)	82 (90.1)
Circulatory Diseases	32,419 (67.7)	30,940 (67.7)	1479 (69.3)	428 (85.4)	218 (86.9)	87 (95.6)
Any Cancer	13,831 (28.9)	13,344 (29.2)	487 (22.8)	164 (32.7)	88 (35.1)	42 (46.2)
Type 2 Diabetes	7841 (16.4)	7409 (16.2)	432 (20.3)	191 (38.1)	107 (42.6)	57 (62.6)
Kidney Diseases	7206 (15.1)	6867 (15)	339 (15.9)	194 (38.7)	119 (47.4)	56 (61.5)
Liver Diseases	4406 (9.2)	4234 (9.3)	172 (8.1)	58 (11.6)	32 (12.7)	14 (15.4)
Autoimmune Diseases	7544 (15.8)	7163 (15.7)	381 (17.9)	109 (21.8)	61 (24.3)	19 (20.9)
Comorbidity score mean (SD)	2.3 (1.5)	2.2 (1.5)	2.3 (1.5)	3.1 (1.6)	3.3 (1.6)	3.9 (1.5)

Abbreviations: BMI, body mass index (calculated as weight in kilograms divided by height in meters squared); COVID-19, coronavirus disease 2019; ICU, intensive care unit; IQR, interquartile range; NDI, 2010 Neighborhood Socioeconomic Disadvantage Index; MM, Michigan Medicine. ^a^ Percentages are reported as fraction of column totals excluding missing entries. ^b^ Includes White Hispanic or unknown; Black Hispanic or unknown; Asian Hispanic, non-Hispanic, or unknown; Native American Hispanic, non-Hispanic, or unknown; Pacific Islander Hispanic, non-Hispanic, or unknown; and other Hispanic, non-Hispanic, or unknown. ^c^ Includes missing race and/or ethnicity.

**Table 2 jcm-10-01351-t002:** Comparison of the top 20 traits from White and Black cohorts across COVID-19 outcome PheWAS.

		Outcome in Top 20 Traits		
Phecode	Description	Hospitalization	ICU Admission	Death
276	Disorders of fluid, electrolyte, and acid-base balance	White	White	Black
276.1	Electrolyte imbalance	White	White	Black
276.12	Hyposmolality and/or hyponatremia	White	White	White
276.13	Hyperpotassemia	White		
276.4	Acid-base balance disorder	Both	Both	Black
276.41	Acidosis	White	Both	Black
276.5	Hypovolemia	White		
401.2	Hypertensive heart and/or renal disease	Both		
427.7	Tachycardia, not otherwise specified (NOS)	White	White	
458	Hypotension	Both	Both	
507	Pleurisy; pleural effusion	White		
508	Pulmonary collapse; interstitial and compensatory emphysema	White	White	
509	Respiratory failure, insufficiency, arrest	Both	Black	Black
509.1	Respiratory failure	Both	Black	Black
509.2	Respiratory insufficiency	Both	Both	Black
509.8	Dependence on respirator [Ventilator] or supplemental oxygen	Both	Black	
585	Renal failure	Both	Both	Black
585.1	Acute renal failure	Both	White	
585.3	Chronic renal failure [CKD]	Both	Both	
586	Other disorders of the kidney and ureters	White		
38	Septicemia	Black	Black	
38.3	Bacteremia	Black	Black	
401.22	Hypertensive chronic kidney disease	Black		
480.2	Viral pneumonia	Black		
506	Empyema and pneumothorax	Black	Black	Black
509.3	Pulmonary insufficiency or respiratory failure following trauma and surgery	Black	Both	Black
585.4	Chronic kidney disease, Stage I or II	Black	Black	
588	Disorders resulting from impaired renal function	Black		
785	Abdominal pain	Black		
994.2	Sepsis	Black		
41.1	Staphylococcus infections		White	
260	Protein-calorie malnutrition		White	
285	Other anemias		White	
287.3	Thrombocytopenia		White	
288	Diseases of white blood cells		White	White
458.9	Hypotension, not otherwise specified (NOS)		Both	
854	Complications of cardiac/vascular device, implant, and graft		White	
276.6	Fluid overload		Black	
411.4	Coronary atherosclerosis		Black	
459	Other disorders of circulatory system		Black	
459.9	Circulatory disease, not elsewhere classifiable (NEC)		Black	
505	Other pulmonary inflammation or edema		Black	
249	Secondary diabetes mellitus			White
250.25	Diabetes type 2 with peripheral circulatory disorders			Both
284	Aplastic anemia			White
284.1	Pancytopenia			White
288.2	Elevated white blood cell count			White
292	Neurological disorders			White
292.1	Aphasia/speech disturbance			White
292.11	Aphasia			White
292.3	Memory loss			White
681.5	Cellulitis and abscess of leg, except foot			White
681.6	Cellulitis and abscess of foot, toe			White
695.9	Unspecified erythematous condition			White
710	Osteomyelitis, periostitis, and other infections involving bone			White
710.1	Osteomyelitis			White
710.11	Acute osteomyelitis			White
710.19	Unspecified osteomyelitis			White
771	Musculoskeletal symptoms referable to limbs			White
962.3	Hormones and synthetic substitutes causing adverse effects in therapeutic use			White
275	Disorders of mineral metabolism			Black
276.11	Hyperosmolality and/or hypernatremia			Black
290	Delirium dementia and amnestic and other cognitive disorders			Black
290.2	Delirium due to conditions classified elsewhere			Black
348	Other conditions of brain			Black
426.21	First degree atrioventricular (AV) block			Black
427.2	Atrial fibrillation and flutter			Black
427.21	Atrial fibrillation			Black
503	Pulmonary congestion and hypostasis			Black

## Data Availability

Data cannot be shared publicly due to patient confidentiality.

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
