# Peer review of "A Phenome-Wide Association Study (PheWAS) of COVID-19 Outcomes by Race Using the Electronic Health Records Data in Michigan Medicine"

_jcm, 2021, doi:10.3390/jcm10071351_

Round 1
Reviewer 1 Report
The paper provides information on a phenome-wide scan to identify pre-existing conditions related to COVID-19 susceptibility and prognosis across the medical phenome and how they vary by race. The authors provide precise and detailed information on comorbidities related to COVID-19 susceptibility and prognosis. The paper would be useful for the need for targeted screening to support specific vulnerable populations to improve disease prevention and healthcare delivery.
General review
The written language of the paper needs a rewrite.
Specific Review
- Introduction
- The content in the introduction section needs to rewrite in more scientific way.
- g., line 60 the sentence” Introduced by Denny et al. in 2010, a phenome-wide association study (PheWAS) is an omnibus scan to identify gene disease associations across the medical phenome [1]. Please rewrite in scientific version.
- The sentence” The main goal of a PheWAS is to replicate known gene disease relationships and to search for hidden and unanticipated associations” had no reference. Please add the reference.
- The sentence” Because COVID-19 is a respiratory disease and produces flu-like symptoms, testing strategies in the US initially focused on those with symptoms, the elderly, and those with pre-existing conditions [8] -populations who are at risk of severe disease and complications” the – should be added after the word symptoms.
- Please add the reference for the sentence “These 73 include liver, kidney, heart, and respiratory disease.”
- Materials and Methods
- The authors in line 98 claimed “constituted our initial study cohort of 53,853 patients, of whom 2,582 tested positive” however in line 167 the authors claimed” Of those eligible for inclusion, our study population comprised 47,862 individuals (ntested=47,862 [npositive=2,133]) who had available International Classification of Disease (ICD; ninth and tenth editions) code data after applying the 14-day-prior to testing restriction to the HER. It is confusing the positive cases were 2582 or 2133?
- Results
- The authors in line 221 claimed “as well 221 as 239 suggestive traits, including 54 circulatory system, 30 respiratory, 29 endocrine/metabolic, and 21 genitourinary diseases” this is confusing. How many traits they study? And how they got this result? What are the numbers?
- The figures are in low quality where the colours are hard to recognize. Please prepare the high quality one with clear colours.
Reviewer 2 Report
The paper by Salvatore et al concerns the impact of pre-existing conditions on COVID-19 outcomes, stratified per ethnicity.
The paper could be of great interest, yet it needs to be revised:
- First and foremost, the method section needs to be made clearer, in some parts it is not possible to understand the number of participants; also the whole section on disparties in testing is quite confusing: it is not clear why the authors didn't select the white and black group so to make them comparable.
- It would be interesting if you discussed a bit more the baseline differences between the two groups in terms of age and sex, given the importance of both variables.
- Also a discussion on medication would be interesting: perhaps the different outcomes may also dffer because of differences in treatment.
- A discussion on the difference of incidence of the different pathologies at baseline between the populations would also help.
- Would it be possible to examine in more detail the different pathologies in the two groups?
- Finally, the language needs to be revised.
